# Memorize and Rank: Enabling Large Language Models for Medical Event Prediction

## Abstract

Medical event prediction produces patient's potential diseases given their visit history. It is personalized yet requires an in-depth understanding of domain knowledge. Existing works integrate clinical knowledge into the prediction with techniques like concept embedding, patient records as knowledge graphs, and external knowledge bases, leaving the knowledge obtained through the pretraining of modern Large Language Models untouched. We introduce MERA, a clinical event prediction model that bridges pertaining natural language knowledge with medical code. We apply contrastive learning on a predicted ranking list for task-specialized optimization. With concept memorization through fine-tuning, we equip the LLM with an in-depth understanding to recall the natural language definitions for medical code during inference. Experimental results on MIMIC datasets show that MERA outperforms state-of-the-art models.

## 1  Introduction

Electronic Health Records, which store patient status and diagnoses made by physicians associated with timestamps, represent valuable domain expertise and clinical operation patterns (Caufield et al. 2019). The diagnosis judgments are made by clinicians based on their medical knowledge, which is largely acquired from years-long education grounded in textbooks and literature, as well as their years of accumulated experience, often implicit and mined in large-scale clinical data. Medical event prediction aims to predict future patient events given their history (Morid, Sheng, and Dunbar 2023). The events are normally presented in medical code format, such as ICD-9 disease codes, with a large candidate space to choose from (13,000 disease candidates in ICD-9) (**?**). A reliable medical event prediction system enables efficiency improvement for hospital operations, early warning of potential diseases for patients (Rochefort, Buckeridge, and Forster 2015), optimized clinical resource and facility allocation (Yadav et al. 2013), and better risk estimation for sustainable insurance (Hsu et al. 2016).

There are two significant challenges for medical event prediction, which have motivated many existing works but have still not been solved. First, what would be the best practice to introduce domain-specific clinical knowledge into the model? Existing works create concept initial embeddings from disease names, or enrich patient representation with external disease ontology. But there is still a significant gap between the primary knowledge modality, *i.e.* natural language, with the model's hidden representation. Second, how can we deal with the large candidate space when producing future event predictions and exploit the supervision signals that could be induced from the candidate space? Existing works mainly consider the task as a $k$-way classification task where $k$ is the number of possible medical events that could happen and then apply binary cross entropy loss for each medical code. The dependencies among candidates and the structure of the medical code ontology are mostly ignored.

Generative Language Models (LM), especially recently introduced Large Language Models (LLM), are trained to predict the next token, follow task instructions, and align with human preference. These models excel in language understanding and reasoning capabilities, as shown in their performance on science-based benchmarks. Additionally, the LLMs absorb a large amount of knowledge mined in literature and online corpus during the pretraining stage. However, there is still an underexplored field of using LLM for medical event prediction due to the aforementioned gap between natural language and medical code and the gap between the token-level optimization process and the outcome-level large candidate output space. These challenges hinder the application of generative LM to diagnosis prediction tasks, while the state-of-the-art models are still graph neural network-based without full utilization of natural language knowledge (Yang et al. 2023; Wu et al. 2023; An et al. 2023). There are works that use transformer-based LM for clinical outcome prediction, but they either have a heavy focus on using natural language medical notes as input (Niu et al. 2024; Wang et al. 2023), or discard the pretrained knowledge (Rasmy et al. 2021; Pang et al. 2021; Li et al. 2023).

To tackle these challenges, we propose MERA, an LLM for medical event prediction with medical code understanding and rich supervision over the output space. The patient's history diagnosis results are formulated as linear sequences and the LLM is expected to produce the probability distribution for the diagnosis result for the next visit. Different from the ordinary decoding process and token-level optimization, we produce the event prediction result from the probabilities distribution of producing corresponding tokens. We apply contrastive learning to force the model to distinguish true diagnosis events from false ones. The contrastive learning pro-

cess is extended to multiple levels in the hierarchical organization of the medical code definitions. The model is learned to distinguish the true events from a pool of potential outcomes while the pool is getting more relevant to the true events. To equip the LM with an understanding of the medical code, we fine-tune the LM to memorize the mapping between medical code and their definitions.

We validate the effectiveness of our method in diagnosis prediction tasks on MIMIC-III and MIMIC-IV datasets. We observe that MERA yields significant improvement to the existing state-of-the-art medical event prediction model.

## 2 Method

### 2.1 Task Formulation

The diagnoses for patients are represented with the ICD-9 code ontology $O$ defined by domain experts, containing medical codes $\{c_1, c_2, ..., c_{|O|}\}$ where $|O|$ is the total number of codes. The codes are organized as a tree structure where the codes are leaf nodes. For each medical code $c$, it is part of the code groups $\{g_1^c, ..., g_{depth(O)}^c\}$ according to its ancestors from the most coarse 1st level to the finest $depth(O)$ level. There is also a one-to-one mapping between the code $c$ and its natural language definition $def_c$. For example, the medical code 250.23 stands for "Diabetes with hyperosmolarity, type I [juvenile type], uncontrolled". It belongs to a top-level group for all "Endocrine, Nutritional, and Metabolic Diseases and Immunity Disorders", and further belongs to the 2nd-level disease group, "Diabetes mellitus" and fine-grained disease group "type I uncontrolled diabetes". Given an electronic health records collection containing medical records for $n$ patients $\{P_1, P_2, ..., P_n\}$, patient history diagnosis can be articulated as a sequence of admission instances $\{V_1^{P_i}, V_2^{P_i}, ..., V_T^{P_i}\}$ for a patient $P_i$ where $T$ is the number of existing visits. For a particular visit $V_j$, the medical judgment made by clinicians as a result of the visit is an unordered set of diagnosis $\{d_1^{V_j}, d_2^{V_j}, ..., d_m^{V_j}\}$ in the format of $m$ unique ICD-9 code ($d \in O$).

For the **diagnosis prediction** task, We aim to predict the diagnosis for the potential next visit $V_{T+1}$ by selecting from the medical code ontology $O$, which can be described as $f_{dp} : \{V_1, V_2, \ldots, V_T\} \rightarrow V_{T+1}$. For the **heart failure prediction** task, which can be described as a binary classification function $f_{hf} : \{V_1, V_2, \ldots, V_T\} \rightarrow 0, 1$, we are more focused and aims to predict whether a patient would encounter heart failure in any of the future visits.

### 2.2 Overview of MERA

MERA takes an existing large language model $LM$ pretrained with a natural language corpus. The model takes the input sequence expressing the patient's history $seq_{history}$ and outputs a probability distribution over the possible next token $P\left(w_1^o \mid seq_{history}\right)$.

There are three steps involved as a pipeline: 1) Training the model to memorize medical codes used to represent the diagnoses; 2) Training the model to learn causal and temporal relations between visits and intra-visit patterns from patient diagnosis; 3) During inference, performing autoregres-

sive generation to produce diagnosis prediction result given an unseen patient history input.

### 2.3 Medical Concept Memorization

To bridge medical codes on $O$ and the natural language knowledge learned through pertaining, we fine-tune $LM$ on synthetic question-answering pairs to create a memorization of the code and natural language definition mapping.

**Bidirectional code and definition memorization.** For each code $c_i$ and the natural language definition $def_{c_i}$, we create two input-output pairs. The first pair prompts the LM with the question "What is the definition of ICD-9 code $c_i$" and the target answer of "$def_{c_i}$" to train the model recall definition given code. The second pair helps the model memorize the mapping reversely by asking "What is the ICD-9 code with the definition of $def_{c_i}$" with an expected answer of "$c_i$". Note that we only apply loss on the completion given the input question while not requiring the model to learn the process of the reconstruction of the code or definition.

**Coding ontology structure memorization.** Besides the short-term memorization, memorization through fine-tuning enables us to embed dependency information among codes in $LM$. For each category level $1..depth(O)$ on the ICD-9 code ontology $O$, the curated pairs map the code to its category, such as ("What is the first-level category of the ICD-9 code 998.51?", "Injury and Poisoning.").

### 2.4 Capturing Inter-visit Relations

The second phase of fine-tuning aims to equip the model with temporal and causal understanding among diagnoses across patient visits. Given a patient history for patient $P_i$, $\{V_1^{P_i}, V_2^{P_i}, ..., V_T^{P_i}\}$, we create a sequence of tokens $Seq_{history}$ to represent the history as input of $LM$. Diagnosis medical codes within a visit $V_i$ are concatenated to form a token segment for a visit and the visit segments are further concatenated with a separator phrase to indicate a new visit.

**Input sequence perturbation.** The order of patient visits is crucial to convey the causal and dependent relation as the diagnosis in a later visit is conditioned on the previous diagnosis. However, the order of diagnosis codes within a particular visit does not matter as they are made simultaneously. An ideal medical event model should preserve the first kind of order while ignoring the second. To achieve this goal with a sequential LM, we propose to create multiple variants of the input patient history sequence. Each variant keeps the same visit order but shuffles the diagnosis code within each visit. By observing the data instances with shuffled order and the same target distribution, we teach the LM to ignore the order of diagnosis code with a model-agnostic design.

**Optimization on ranking.** After observing the input sequence of patient history, $LM$ is expected to output the probability distribution of the first diagnosis code $P\left(w_1^o \mid seq_{history}\right)$ of the upcoming visit $V_{T+1}$. Ordinary language modeling applies cross-entropy loss on the probability of predicting the correct next token. We further provide fine-grained supervision on the probability distribution of the new token. The code candidate space includes

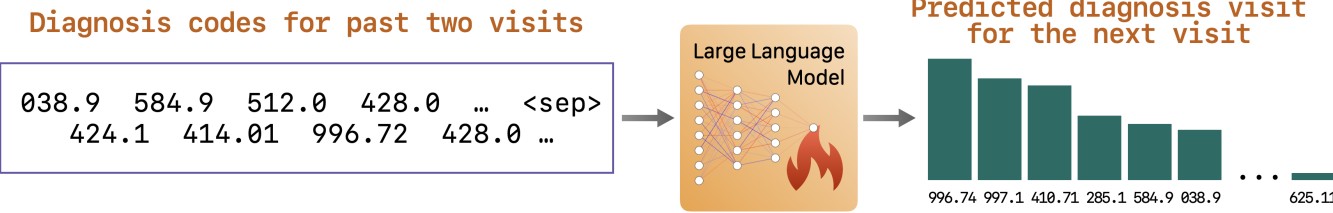

Figure 1: The model design of MERA.

$\{c_1, c_2, ..., c_{|O|}\}$. There is $|pos|$ diseases that do happen in the next visit $\{c_1^{pos}, c_2^{pos}, ..., c_{|pos|}^{pos}\}$ and $|neg|$ diagnosis that is not included in the next visit $\{c_1^{neg}, c_2^{neg}, ..., c_{|neg|}^{neg}\}$ among all code candidates ($|pos| + |neg| = |O|$). We obtain the probability of predicting each code candidate as the next token $P(w_1^o \mid seq_{history}) = \{p_1, p_2, ..., p_{|O|}\}$ by applying a softmax over the logits of all candidate codes. We then apply the training objective to the probability distribution.

**Label space-driven hierarchical contrastive learning.** We design training objectives to equip the model to identify the positive diagnosis among a group of candidate diagnoses sharing similar properties. With such a design, the model is forced to learn the subtle differences among similar diseases and specifically optimize itself for the diagnosis capability for various granularity of the decision space. For a particular $i$-th category level on the output candidate space $O$, we identify positive diagnosis codes that do appear in the next visit. We then apply InfoNCE loss (Oord, Li, and Vinyals 2018; Ma et al. 2021; Meng et al. 2021) shown below to parameterize the model to identify the correct diagnosis among all the same-category candidates. The produced loss for each subgroup is then added across different subgroups of the same ontology level and then further aggregated by the sum over different ontology levels.

### 2.5 Modeling Intra-visit Dependencies

Besides training the model to reason between visits, there are many implicit, unspoken rules and dependencies mined in the large pool of diagnoses within each visit. For example, among the same group of similar diseases, the clinician normally only chooses the most representative code for the patient's status; some diseases might suppress or correlate with the appearance of other diseases. Without modeling the intra-visit dependencies, we ignore real-life clinic operation patterns, which could lead the model to yield similar but unrealistic diagnosis predictions. The diagnosis prediction made for a specific visit should consider other diagnoses for the same visit. To model the intra-visit dependencies, we create multiple teaching forcing training instances, each including a segment of the target diagnosis sequence to simulate the diagnosis process that partial diagnosis decisions are provided. For each instance, we include $k$ diagnosis in the target diagnosis list $V_{T+1}$ where $k \in \{0, .., |V_{T+1}|\}$ in the end of the input sequence $seq_{history}$. We then apply contrastive learning over the ranking list of the probability distribution of the next output token $P(w_{k+1}^o \mid seq_{history} : w_1^o, ..., w_k^o)$.

For the positive diagnosis code for the $k + 1$-th output diagnosis token, we consider all diagnoses in $V_{T+1}$ except $w_1^o, ..., w_k^o$ as those diagnoses have been predicted in previous output steps.

### 2.6 Autoregressive Decoding during Inference

After the two fine-tuning stages, learning the mapping between medical code and natural language knowledge and learning inferring diagnosis, the produced $LM$ can be used to perform inference for unseen patient history. Given the $seq_{history}$, $LM$ performs autoregressive decoding to output the discrete diagnosis label with the highest probability in the ranking list for each output step until the end-of-sentence token is generated.

## 3 Experiments

### 3.1 Experimental Setup

**Datasets.** We use MIMIC-III (Johnson et al. 2016) and MIMIC-IV (Johnson et al. 2023) EHR datasets containing patient records to evaluate the effectiveness of MERA. They are large, de-identified, and publicly available collections of medical records collected at the Beth Israel Deaconess Medical Center in Boston, Massachusetts, USA. The MIMIC-III dataset focuses on the patients who are eventually admitted to ICU, and the MIMIC-IV dataset includes both ICU patients and other patients. We conduct data preprocessing following previous works (Yang et al. 2023; Lu, Han, and Ning 2022). We keep the patients with more than one visit in the datasets. We use the full ICD-9 code (instead of a simplified and higher-level code, which makes the task easier) to represent the diagnosis. To avoid data duplication during the overlapping time range between the two datasets, only the information of patients from MIMIC-IV with multiple visits between 2013 and 2019 was used. They are randomly split based on patients into training, validation and testing sets. For MIMIC-III and IV, the training, validation, and test sets contain 6000, 493, and 1000 patients, and 8000, 1000, and 1000 patients respectively. For patient history, we use the last visit as the label, while the earlier visit is input.

**Evaluation metrics.** For diagnosis prediction, we report the weighted F1 score and Recall@k metrics, where k is the number of top-ranked predicted diseases to consider. The weighted F1 score measures the accuracy of disease prediction by calculating the harmonic mean of precision and recall. They are both higher, the better. For heart failure prediction, we report AUC, which measures the area under the

| # | Model | Diagnosis Prediction | | | | | | Heart Failure | | | |
|---|---|---|---|---|---|---|---|---|---|---|---|
| | | MIMIC-III | | | MIMIC-IV | | | MIMIC-III | | MIMIC-IV | |
| | | w-F1 | R@10 | R@20 | w-F1 | R@10 | R@20 | AUC | F1 | AUC | F1 |
| *RNN/CNN and attention-based models* | | | | | | | | | | | |
| 1 | Deepr | 18.87 | 24.74 | 33.47 | 24.08 | 26.29 | 33.93 | 81.36 | 69.54 | 88.43 | 61.36 |
| 2 | Dipole | 19.35 | 24.98 | 34.02 | 23.69 | 27.38 | 35.48 | 82.08 | 70.35 | 88.69 | 66.22 |
| 3 | Timeline | 20.46 | 25.75 | 34.83 | 25.26 | 29.00 | 37.13 | 82.34 | 71.03 | 87.53 | 66.07 |
| 4 | RETAIN | 20.69 | 26.13 | 35.08 | 24.71 | 28.02 | 34.46 | 83.21 | 71.32 | 89.02 | 67.38 |
| 5 | HiTANet | 21.15 | 26.02 | 35.97 | 24,92 | 27.45 | 36.37 | 82.77 | 71.93 | 88.10 | 68.21 |
| *Graph-based models* | | | | | | | | | | | |
| 6 | G-BERT | 19.88 | 25.86 | 35.31 | 24.49 | 27.16 | 35.86 | 81.50 | 71.18 | 87.26 | 68.04 |
| 7 | GRAM | 21.52 | 26.51 | 35.80 | 23.50 | 27.29 | 36.36 | 83.55 | 71.78 | 89.61 | 68.94 |
| 8 | CGL | 21.92 | 26.64 | 36.72 | 25.41 | 28.52 | 37.15 | 84.19 | 71.77 | 89.05 | 69.36 |
| 9 | MCDP | - | 28.30 | 39.60 | - | 25.80 | 36.10 | - | - | - | - |
| 10 | Chet | 22.63 | 28.64 | 37.87 | 26.35 | 30.28 | 38.69 | 86.14 | 73.08 | 90.83 | 71.14 |
| 11 | KGxDP | 27.35 | 30.98 | 41.29 | 30.38 | 34.19 | 43.47 | 86.57 | 74.74 | 95.66 | 79.87 |
| *Fine-tuned Transformer-based models* | | | | | | | | | | | |
| 12 | MERA (LLaMA-7B) | **32.77** | **35.94** | **47.48** | **39.26** | **38.77** | **51.61** | **89.49** | **77.21** | **97.26** | **82.31** |

Table 1: Performance comparison with baselines (%).

| Model | Code Acc | Definition Acc |
|---|---|---|
| MERA (LLaMA-7B) | 99% | 91% |

Table 2: Evaluation of the memorization capabilities for ICD-9 codes.

receiver operating characteristic curve, and F1 score, which evaluates the balance between precision and recall.

**Baselines.** *RNN/CNN and attention-based models:* RE-TAIN (Choi et al. 2016) employs two attention mechanisms to model two-way visit-disease mapping. Dipole (Ma et al. 2017) proposes a bidirectional RNN to address the issue of lengthy medical visit records. Timeline (Bai et al. 2018) designs an attention mechanism that combines time intervals and attention weights of each entity. HiTANet (Luo et al. 2020) employs a hierarchical temporal attention mechanism. Deepr (Nguyen et al. 2017) predicts future risks from medical records by converting records into discrete element sequences and using a CNN to detect predictive local clinical patterns. *Graph-based models:* GRAM (Choi et al. 2017) employs the structure of medical ontologies. G-BERT (Shang et al. 2019) integrates pretrained language models and fully considers the hierarchical information found in ICD-9 codes. CGL (Lu et al. 2021) introduces a collaborative graph learning model. Chet (Lu, Han, and Ning 2022) computes the diagnosis neighbor and global neighbor for each disease. MCDP (Li and Gao 2022) presents a methodology that uses hyperbolic space and multi-modal contrastive loss to preserve the hierarchical structure of diagnostic codes. KGxDP (Yang et al. 2023) formulates each patient as a personalized medical KG, combining medical KGs with patient admission history.

## 3.2 Performance of Diagnosis Prediction

We show the performance on the diagnosis prediction task in Table 1. We observe that the graph-based models, in general, yield better performance compared with the RNN/CNN-based sequential model. The best-performing baseline would be the KGxDP model, which uses GNN to introduce spatial features and utilizes G-BERT pretraining knowledge to initialize the initial representation of the GNN.

We observe that our proposed model MERA achieves significantly better performance for both diagnosis prediction and heart failure prediction tasks for both datasets. There is an almost 9-point higher weighted F1 score and more than 8-point higher recall@20 for MIMIC-IV.

## 3.3 Performance of Memorization

We further evaluate whether the trained model, after the medical concept memorization fine-tuning, can recall the medical code given definition sentences or recall the definition sentences given the ICD-9 code. We report code accuracy and definition accuracy in Table 2, and the model has to produce the exact same code and definition to count as a hit. We observe that the LM can remember the code-definition mapping almost perfectly, indicating the effectiveness of our proposed memorization technique.

## 4 Conclusion

By integrating domain-specific clinical knowledge and addressing the complexities of a large candidate space, MERA bridges the gap between natural language processing and medical code understanding. Our rigorous validation of MIMIC datasets has established MERA as a leading approach for medical event prediction.

## Ethical Considerations

The trained diagnosis prediction model inherits bias from multiple sources, including pre-training corpus, medical records distribution used for fine-tuning and more. The model should be fully evaluated before it is considered to be deployed in real clinical scenarios. The outcome of the diagnosis prediction model should not be used to serve as a factor to trace the discrimination label for specific diseases. Hospitals and insurance companies should not use the predicted diagnosis as the reason or motivation to change their service to patients.

## Reproducibility Statement

Our code will be released along with the published paper.

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
