# OpenReview forum: "Memorize and Rank: Enabling Large Language Models for Medical Event Prediction"
_AAAI.org/2024/Spring_Symposium_Series/Clinical_FMs — AAAI 2024 SSS on Clinical FMs_

### Official Review · Reviewer_Ls2v · 2024-02-20
**Good Adaptation of LLMs for Medical Event Prediction with SOTA Performance**

**Rating:** 7
**Confidence:** 4

**Review:**

### Summary

This paper proposes a new strategy to solve “medical event prediction”. They motivate the problem, describe its main challenges and propose a solution that addresses these challenges. The proposed method consists of two main steps - (i) medical concept memorization - to adapt the vocabulary of the LLM, and (ii) contrastive learning to capture inter and intra-visit relations in medical events. This method achieves SOTA performance on MIMIC-III and MIMIC-IV datasets.

### Strengths:
- Provides a clear motivation for the problem and outlines the main challenges in solving the problem.
- Novel and effective LLM pre-training tasks are proposed to directly address the above challenges
- A diverse set of baselines are considered is evaluating the proposed method

### Weaknesses:
- No ablation study in experiments to understand which of the pretraining tasks leads to performance gains.
- Does not consider any Clinical LMs as a base model in experimentation. Clinical LMs like BioGPT, MedPaLM may already understand medical vocabulary. This may remove the need for the “memorize” phase in the proposed method.
- Poor readability in certain sections. Eg: explanation of proposed method can be improved by correcting grammatical errors and adding equations/figures.

### Other Feedback:
- Other baselines also evaluate on MIMIC-III and MIMIC-IV. Consider releasing your train/test split and characteristics of the dataset in your next iteration of this work. This could help standardize research in the “medical event prediction” field.
- Since your method is built on top of a generative LM, it could be inherently better than other methods at quickly (with limited data) understanding new medical events. It would be interesting to see experiments on predicting medical events in a few-shot/zero-shot setting.
- The authors noted that some other methods modeled external information as knowledge graphs. There are some advantages to these methods (eg: updating information) and those should be further explored in future work. For example, authors could add pre-training tasks using the knowledge graphs, similar to the way the ontology of medical codes is currently used.

---

### Official Review · Reviewer_xarc · 2024-02-21
**Foundation model for medical code prediction by code - definition training.**

**Rating:** 6
**Confidence:** 4

**Review:**

**Summary**
The paper proposes a large language model-based foundation model for medical code prediction tasks. The proposed model is based on LLaMA-2-7B, with further fine-tuning on (1) the medical code and text definition pretext task; (2) the medical code prediction task. The model is tested on MIMIC-III and MIMIC-IV, with a comparison to RNN/CNN-based models and graph-based models.

**Strengths**
- Clear motivation for the "concept memorization" training and "hierarchical contrastive learning".
- I appreciate that the authors include a task formulation section, for readers to better understand the task and data format.
  - This part can be further improved if authors can clarify whether ``diagnosis prediction`` is a multi-label classification
- The technical part is easy to follow

**Weaknesses**
- The input sequence perturbation makes sense to me. However, authors do not explicitly mention whether the within-visit order of target visit matters or not.
  - During training, will the sequence perturbation also apply to the target visit?
  - During evaluation, how the accuracy/precision/recall are calculated? (If target code order is ``996.74, 428.0, 414.1``, and model outputs ``414.1, 428.0, 996.74``)
- Authors mention that transformer-based LMs are widely used in the same task (Paragraph#3 in the Introduction Section). Is there a reason transformers are not included as compared baselines?

**Misc.**
- Missing reference in Paragraph#1 in the Introduction Section.
  > 13,000 disease candidates in ICD-9 ``(?)``

Overall, the quality of this paper is great. I would be very interested in seeing authors' responses regarding my concerns.

---

### Official Review · Reviewer_Kwhx · 2024-02-26
**good paper solving important problem of 'next' medical disease prediction**

**Rating:** 6
**Confidence:** 3

**Review:**

The authors present an approach to predicting the next most likely disease /outcome for a patient given a previous sequence of outcomes. They focus on using the icd10 ontology. The model is an LLM that is further fine tuned on intermediate predictive tasks such as recalling definitions of codes. The authors trial their approach on open source EHR data and show good performance. The paper is a good contribution to an important area of research.

In general I have two criticism to offer.  My criticism is on the practicality of using approach in the real world. Based on my reading it assumes codes are accurately recorded by the clinician/clinical care team following a given patient episode. In reality clinicians are not very good at recording this level of data. At what stage of the clinical workflow would this 'model' would this be useful?

Secondly I am not sure how much is lost by discarding the actual clinical text as part of the input data. This approach seems to model the problem as a sequence of codes (or sequence of bag-of-codes). LLMs are designed to interpret free text so why not take advantage of this?

Overall I think the paper presents an interesting and somewhat original solution to the problem. I think the clarity of writing can be improved and ideally some justification for design choices needed to be provided.

---

### Official Review · Reviewer_kvo1 · 2024-02-27
**Memorize and Rank: Enabling Large Language Models for Medical Event Prediction**

**Rating:** 6
**Confidence:** 4

**Review:**

Authors present an approach to use language model to predict the diagnosis of the next visit of patients given as input a sequence of visits and their associated code-diagnose. They benchmark the new approach with existing approaches and standard datasets form medical settings.

The paper is original in the use of language model technology for next visit prediction, however the reviewer and reader would appreciate more detail in how the system is built and how it learns, specific parameters. What type of model do authors use, what is the architecture, is it an encoder-decoder, only decoder architecture? How is the fine tuning performed?

 I would like to suggest that the clarity of the writing could be improved. There are some sections where the language appears a bit unclear or could benefit from a revision for smoother readability. I recommend a thorough review of the English language throughout the manuscript to enhance its overall clarity. This will undoubtedly contribute to a better understanding of the valuable research.

---

### Official Review · Reviewer_RGj5 · 2024-02-28
**The work introduces a new approach to the field of clinical outcome prediction by using medical coding as input for a transformer-based LM model. The paper primarily focuses on taking a pretrained language model (LM) and fine-tuning it to memorize medical codes, initially to represent diagnoses, and then to learn both causal and temporal relations between visits as well as within visits. The work demonstrates superior performance in two types of prediction tasks across two datasets.**

**Rating:** 4
**Confidence:** 3

**Review:**

Strength:
The method outperfomrs state-of-the-art models: the method yields significantly better performance on diagnois production and heart failture prediction for two MIMIC datasets.


Weak points and suggestions:
1. Missing important related work on bridging ICD code with LLM:
(1) "DRG-LLaMA : tuning LLaMA model to predict diagnosis-related group for hospitalized patients"
(2) "CPLLM: Clinical Prediction with Large Language Models"
2. Unclear writing:
(1) This sentence is confusing to read: “But there is still a significant gap between the primary language, i.e., natural language, with the model’s hidden represntation”
(2) In the 'Performance of Memorization' section, what is the sample size? How many ICD-9 codes were assessed?
3. Presentation error:
(1) Missing citation: "The events are normally presented in medical code format, such as ICD-9 disease codes, with a large candidate space to choose from (13,000 disease candidates in ICD-9) (?)"
(2) Figure 1 is difficult to read and could benefit from additional labeling.